# Study of Wax Deposition Pattern of High Wax-Bearing Crude Oil Based on Cold Finger Experiment

**Lixin Wei** [1,*]**, Da Li** [1]**, Chao Liu** [1]**, Zhaojun He** [2] **and Yang Ge** [1]

[1] Key Laboratory of Enhanced Oil Recovery, Northeast Petroleum University, Ministry of Education, Daqing 163318, China; ld13159815585@163.com (D.L.); nepuliuchao@126.com (C.L.); nepuld@126.com (Y.G.)

[2] Second Oil Production Plant, Daqing Oilfield Company Limited, Daqing 163414, China; nepuhzj@163.com

\* Correspondence: weilixin73@163.com or weilixin@nepu.edu.cn

**Abstract:** In order to solve the problem of wax deposition in waxy crude oil from the Daqing oilfield, cold fingers were used in the experimentation. Compared with other methods, the cold finger method is simple, easy to operate, and takes little space. Measurements of wax deposition with temperature, temperature differences between the crude oil and the wall, deposition time, and cold finger rotation rate were made. The results showed that the deposition rate is up to 0.35 g/h at 8–24 h. The maximum deposition rate at 90 rotations/min was 0.26 g/h, which is 3% higher than the minimum deposition rate.

**Keywords:** deposition pattern; cold finger experiment; crude oil with wax; oil pipeline

## 1. Introduction

Among today's crude oil transportation methods, pipeline transportation is widely used in practice because of its advantages such as economic and environmental protection, large transmission volume, and convenient on-site management. In the process of transportation, due to the existence of temperature difference and pressure change factors in the oil wall, wax, asphaltene, and other components in the oil well gradually precipitate and attach to the pipe wall to form wax deposits [1], leading to the emergence of a reduced flow cross-sectional area of the pipeline, increased resistance, weakened transport capacity, and even the shutdown of production, which increases the difficulty of transporting crude oil containing wax and the risk of pipeline operation [2,3]. At present, nearly more than 80% of crude oil output from oil fields in China is high wax-containing crude oil. Wax-containing crude oil from the seafloor has been gradually developed in recent years, and it is foreseeable that wax-containing crude oil will become an indispensable part of China's oil and gas resources in the future [4], so that the wax deposition pattern of crude oil can be effectively grasped and maintenance measures such as regular pipe cleaning can be carried out [5,6].

To address this problem, domestic and foreign scholars have conducted experiments to simulate crude oil in pipelines to analyze its laws [7]. Hamouda et al. [8] improved the cold plate method apparatus to the cold finger method experimental apparatus, whose experimental principle is basically the same as the cold plate method [9] but can better simulate the wax deposition state in the flow state, and the cold finger method was used for the experimental research [10]. Fan Kaifeng et al. concluded that wax molecular diffusion plays a crucial role in the deposition process with cold finger experiments [11,12]. Majeed, Agrawal, Hamouda, Riberio, Brown, and Huang Qiyu et al. chose the ring channel method for their simulation experiments [13,14]. Due to the high cost of the experimental loop, the experimental cost is large, and the thickness of the sediment cannot be directly obtained. In summary, in this paper, a cold finger-type experimental setup is used which is simple and economical and ensures the simulation of wax deposition in the tubular flow condition, and thus the better study of the wax deposition principles.

## 2. Materials and Methods

### 2.1. Experimental Setup

The experimental setup for this experiment consisted of a cold finger, an oil tank, two circulating water bath controllers, and an additional stirring paddle. A schematic diagram of the experimental setup is shown in Figure 1.

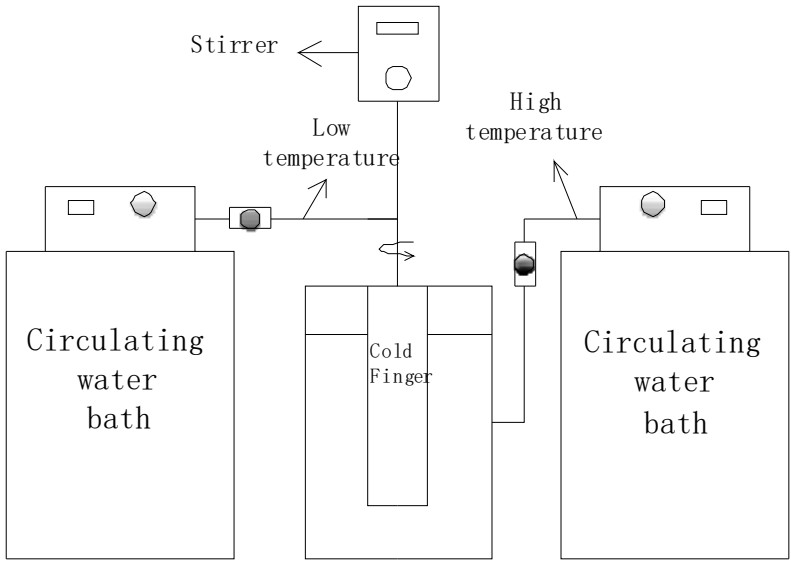

**Figure 1.** Schematic diagram of the device of cold finger.

Two of the circulating water bath controllers control the temperature of the oil in the tank and the wall of the cold finger, respectively. After setting the temperature after turning on the water bath controller, the water in the water bath pot is heated and circulated in the middle of the controller, and the wall of the cold finger and the sandwich of the tank, respectively, while exchanging heat to provide a constant temperature to the wall of the cold finger and the oil flow in the tank. The stirring paddle can be placed into the tank through a hollow cylinder in the middle of the cold finger cylinder, and the stirring paddle is equipped with a motor transmission that can be adjusted to change the shear conditions in the tank to simulate the flow of oil in the pipeline.

### 2.2. Experimental Medium

The experimental medium is a wax-containing crude oil from an oil reservoir in the Daqing oilfield, whose physical parameters are 31 °C freezing point, 855.7 kg/m$^3$ density at 20 °C, 17.15% wax mass fraction, and 49.8 °C waxing point, respectively.

### 2.3. Experimental Steps

Firstly, the crude oil was pretreated by heating the water bath at 60 °C for 6 h, then the experimental temperature of the circulating water bath controller was set, and the oil sample was added after the temperature reached the experimental requirement, and the RW20 stirrer, produced by IKA, Germany, was turned on to adjust the rotational speed to the required speed for the experiment. Turn off the stirrer after the deposition time reaches 48 h, open the valve of the lower part of the tank to make the oil sample flow out inside the tank, leave it for 10 min, remove the cold finger from the tank after all the oil inside the tank is emptied, scrape the deposition from the surface of the cold finger using a tool, and put it into the sampling box for weighing. Since there may be human error in the experiment, each test was repeated twice to take the average value as the final experimental result for subsequent analysis and comparison.

## 3. Result and Discussion

### 3.1. Oil Temperature

(1) The cold finger wall temperature was set to 39 °C and the asphaltene content of the oil to 4.5 wt.%, without turning on the stirring paddle motor, then the tank wall temperature was gradually increased to 41 °C, 43 °C, 45 °C, 47 °C, and 49 °C, respectively, at which the experiment was conducted, and the results were weighed twice to take the average value.

(2) The tank wall temperature was set to 51 °C and the oil asphaltene content to 4.5 wt.%, without turning on the stirring paddle motor, the cold finger wall temperature was gradually increased, respectively, to 41 °C, 43 °C, 45 °C, 47 °C, and 49 °C as with the above method so to take the average value of the two sets of data, as recorded in Figure 2.

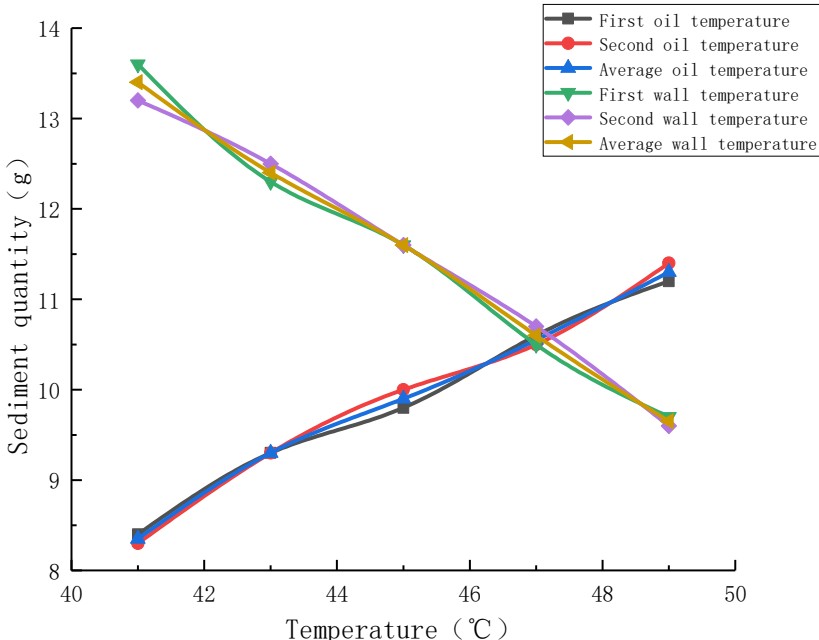

**Figure 2.** Sediment mass curves at the temperatures of different oils and walls.

From the oil temperature curve in Figure 2, it can be seen that the sludge mass monotonically increases when the wall temperature is kept constant, thus gradually increasing the oil temperature. When the oil temperature increases, the temperature gradient at the cold finger wall also increases, and the oil sample in the tank rises with the temperature of the pipe wall. The temperature gradient at the cold finger wall increases as the oil temperature increases, and the diffusion rate of the wax molecules increases accordingly [15]. The wax deposition rate was maximum at 0.24 g/h at 49 °C, and the maximum deposition was 35.4% higher than the minimum.

From the wall temperature curve in Figure 2, it can be seen that by gradually increasing the wall temperature while keeping the oil temperature constant, the sludge mass gradually decreases after the initial maximum, i.e., the temperature difference increases as the sludge mass increases. In this experiment, the increase of the oil wall temperature difference is beneficial to the deposition of the sludge, but when the cold finger surface temperature is too low, it will increase the viscosity of the oil sample around the cold finger surface, and this increase of viscosity will reduce the sludge mass. That is, the larger oil wall temperature difference and the lower cold finger surface temperature have an increasing and weakening effect on the deposition of the sludge at the same time. From the results of this experiment, we can know that the sludge mass decreases with the temperature difference under this experimental condition. The maximum deposition rate was 0.56 g/h at 41 °C, which was 38.9% higher than the lowest deposition rate.

### 3.2. Temperature Interval

The cold finger wall temperature and oil tank wall temperature difference was set at a constant of 10 °C, with the oil asphaltene content of 4.5 wt.%, without turning on the mixer motor, and the temperature difference and temperature interval with the experiment on it were gradually increased, and the sludge obtained from each test was weighed to derive its mass. The mean values were taken as described above and recorded in Figure 3.

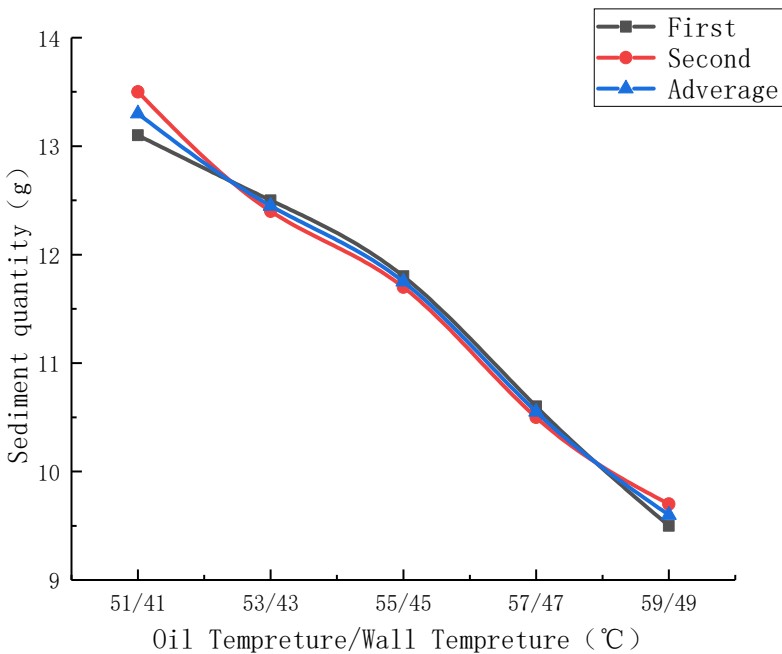

**Figure 3.** Sediment quality curves at the ranges of different temperatures.

As can be seen from Figure 3, the sludge mass reaches a maximum at the initial oil temperature and gradually decreases as the oil wall temperature interval gradually increases. This is because, under the same temperature difference, the higher the oil temperature, the greater the dissolution effect of the oil in the tank on the sludge, which makes the already precipitated wax dissolve again so that the sludge mass shows a decreasing phenomenon with the increase of the oil wall temperature range. The deposition rate was 0.58 g/h at the maximum oil temperature of 51 °C and the wall temperature of 41 °C, and the deposition amount was 38.5% higher than the lowest.

### 3.3. Deposition Time

The tank wall temperature was set at 55 °C, the cold finger wall temperature was 45 °C, the asphaltene content of the oil was 4.5 wt.%, and the deposition time of the oil was gradually increased, without turning on the mixer motor, to 6 h, 12 h, 18 h, 24 h, 30 h, 36 h, 42 h, and 48 h, respectively, at which the experiments were conducted, and the sludge obtained from each test was weighed to give its mass. The mean values were taken as described above and recorded in Figure 4.

As can be seen from Figure 4, with the gradual increase of the deposition time, the mass of the sediment gradually increased and reached the maximum at 48 h. In the first 12 h, the sludge mass growth rate was slow; at 18–24 h, the sludge mass growth rate was higher, to 0.35 g/h after a period of deposition; at about 42–48 h, the deposition amount tended to stabilize, where at this time the wax in the oil was basically all precipitation, with the highest deposition rate being 45.8% higher than the average rate. As the deposition time increases, the thickness of the wax deposition also increases, which weakens the ability of the crude oil to dissipate heat to the outside, thus resulting in a reduction in the temperature difference between the oil walls, so that the amount of wax deposition tends to level off [16].

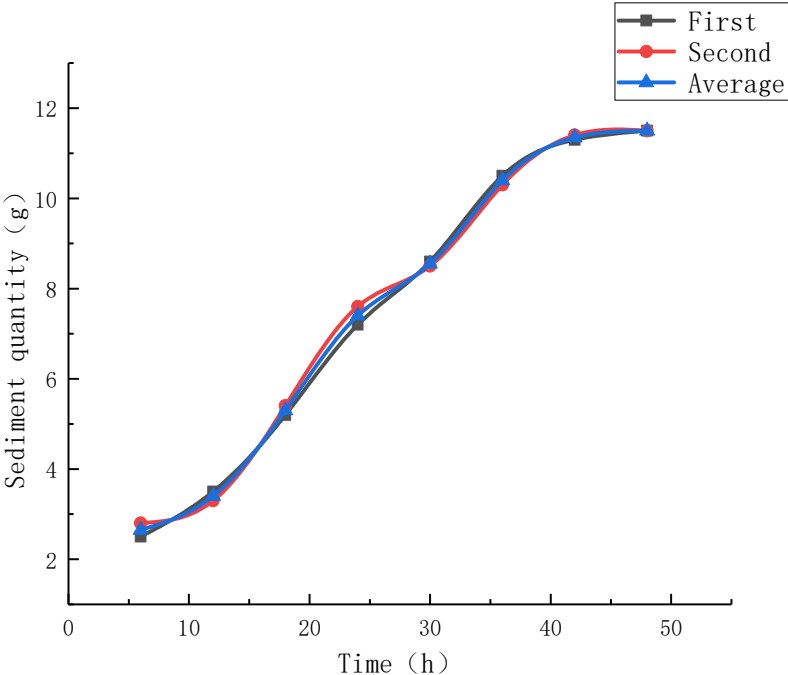

**Figure 4.** Sediment quality curves at the time of different depositions.

### 3.4. Shear Strength

The tank wall temperature was set at 55 °C, the cold finger wall temperature was 45 °C, the asphaltene content of the oil was 4.5 wt.%, and the deposition time was 48 h. The mixing paddle speed was gradually increased to 0 r/min, 30 r/min, 60 r/min, 90 r/min, and 120 r/min for the experiments, and the sludge obtained from each test was weighed to give its mass. The mean values were taken as described above and recorded in Figure 5.

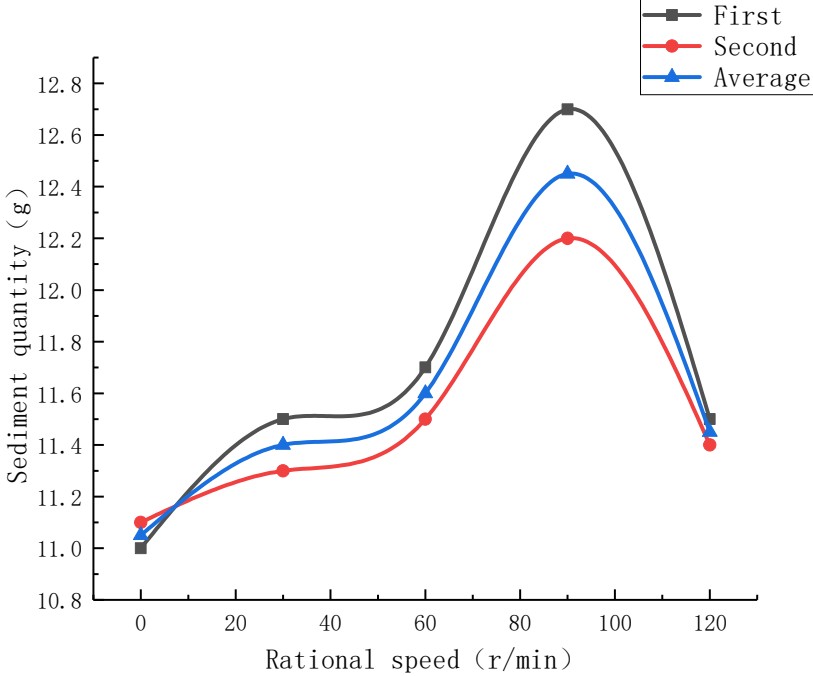

**Figure 5.** Sediment quality curves at different shear rates.

From Figure 5, it can be seen that when the rotational speed was below 90 r/min, the sediment mass increased with the gradual increase of the rotational speed, and reached the

maximum at the rotational speed of 90 r/min when the deposition rate was 0.26 g/h. When the rotational speed was higher than 90 r/min, the rotational speed continued to increase, and the sediment mass gradually decreased, where the maximum deposition increased by 3% compared to the lowest. The reason for this phenomenon is that the rotational speed can increase the temperature gradient and increase the diffusion rate of the wax molecules, which increases the sludge mass [17], but the large shear rate provided by too high of a rotational speed can also have a scouring effect on the sludge that precipitated and adhered to the cold finger surface. When the rotational speed was less than 90 r/min, and when the rotational speed provided less shear stress, the effect of the increased temperature gradient leading to increased sludge mass was more significant, so the sludge mass increased. When the rotational speed was greater than 90 r/min, the shear stress on the surface of the cold finger was greater, and the scouring effect on the sediment on the surface of the cold finger was significantly greater, so the sediment mass decreased with the increasing rotational speed at this time [18].

### 3.5. Oil Repellent Content

The tank wall temperature was set at 55 °C, the cold finger wall temperature was 45 °C, the asphaltene content of the oil was 4.5 wt.%, and the deposition time was 48 h. The oil repellant content was gradually increased to 0.5%, 0.8%, 1.1%, 1.4%, and 1.7% without turning on the stirring paddle motor, and the sludge obtained from each test was weighed to obtain its mass. The oil repellent added in this experiment was a polyacrylamide (PAM) oil repellent with the molecular formula, and the mean values were taken as described above and recorded in Figure 6.

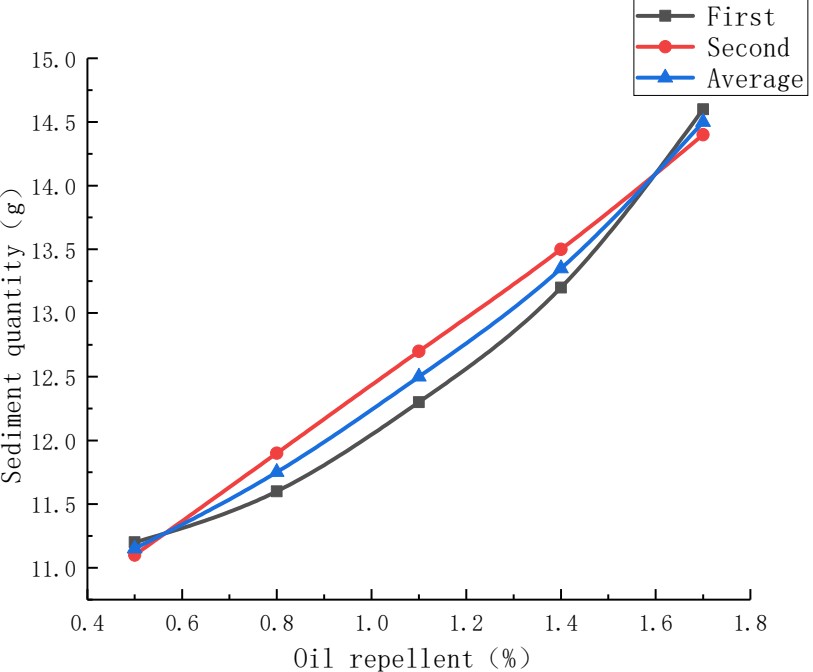

**Figure 6.** Sediment quality curves at different oil displacement agent content.

As can be seen from Figure 6, the mass of sludge increased with the increase of the oil repellent content and reached the maximum when the oil repellent content increased to 1.7%, wherein the deposition rate was 0.6 g/h at this time, and the maximum deposition volume was 30% higher than the lowest. The reason for this phenomenon is that the oil repellent can wrap some materials in the oil to form crystalline cores, which will increase the deposition amount of the sediment. However, at the same time, the oil repellent will work together with the wax molecules to suppress the deposition rate of the sediment,

and it can be seen from the above graph that, in this experimental condition, the effect of increasing the sediment yield was greater than the effect of suppressing the growth rate.

### 3.6. Asphaltene Content

Setting the tank wall temperature at 55 °C, the cold finger wall temperature at 45 °C, and the deposition time at 48 h, the asphaltene content of the oil was gradually increased without turning on the mixer motor to 3.5%, 4.0%, 4.5%, 5.0%, and 5.5%, respectively, at which the experiments were conducted, and the sludge obtained from each test was weighed to give its mass. The mean values were taken as described above and recorded in Figure 7.

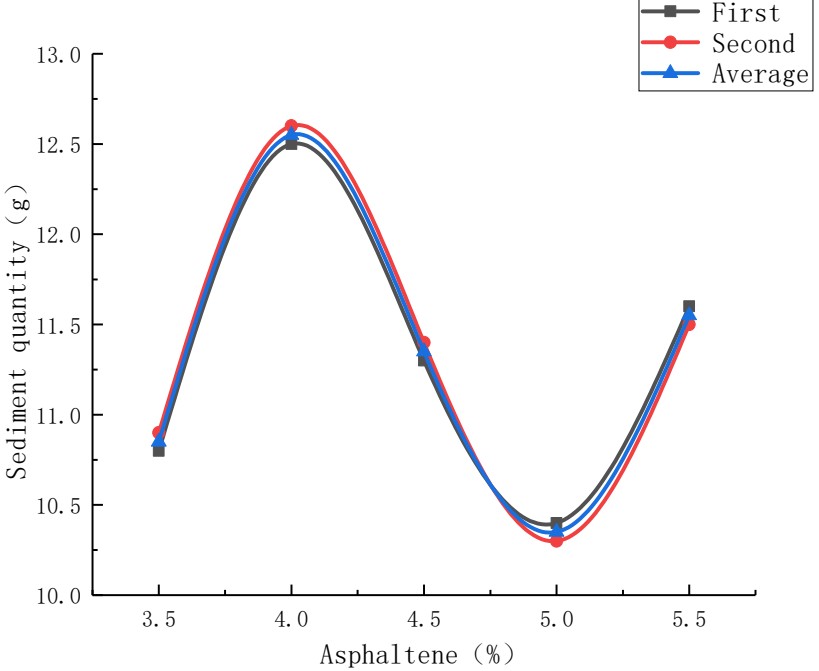

**Figure 7.** Sediment quality curves at different asphaltene content.

As can be seen from Figure 7, the sediment mass showed a wavy trend of increasing, then decreasing, and then increasing with increasing asphaltene content, reaching a maximum at 4% asphaltene content when the deposition rate was 0.26 g/h. Then, when continuing to add asphaltene to 5% when the sediment mass dropped to a minimum at the deposition rate of 0.22 g/h, the sediment mass increased again with an increasing asphaltene content. The highest deposition was 21.3% higher than the lowest. This is due to the fact that when the asphaltene content of the oil is low, the main reason for the waxing of the pipe wall being the migration of the wax molecules, the sludge mass gradually increased at this time. When the asphaltene content exceeded a certain value, the light component in the oil was relatively reduced, the viscosity of the oil increased, the migration of wax molecules was blocked, and the sludge mass gradually decreased as the asphaltene concentration increased. When the asphaltene content continued to increase, the main cause of the wax deposition changed from the migration of the wax molecules to the viscosity-enhancing effect of the asphaltene, due to the excess of asphaltene at this time, and the sludge mass increased again with the increasing asphaltene content [19].

### 4. Conclusions

In the actual transport of crude oil, attention should be paid to the temperature difference between the transported crude oil and the pipe wall so to control it in a relatively reasonable range, to control the deposition time and the shear stress of the crude oil, as

well as the internal asphaltene content and other components, all of which will have an impact on the wax deposition of the crude oil.

**Author Contributions:** Writing—review and editing, L.W.; writing—original draft preparation, D.L.; Conceptualization, C.L.; resources, Z.H.; data curation, Y.G. All authors have read and agreed to the published version of the manuscript.

**Funding:** This research was funded by [Natural Scidence Foundation of Shandong Province for Youth] grant number [ZR2020QE111] And The APC was funded by [Postdocroral Science Foundation of China].

**Conflicts of Interest:** The authors declare no conflict of interest.

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
