# Peer review of "Study of Wax Deposition Pattern of High Wax-Bearing Crude Oil Based on Cold Finger Experiment"

_processes, doi:10.3390/pr10010103_

Round 1

Reviewer 1 Report

In the manuscript Study of Wax deposition pattern of high wax-bearing crude oil based on cold finger experiment authors present several data highlighting the role of several variables, including oil temperature, temperature interval, deposition time, shear strength, oil-repellent content, and asphaltene content on the amount of wax deposited in a cold finger setup.

Having reviewed the manuscript, it is my recommendation that the current manuscript be rejected as it does not further our understanding of wax deposition. Here are the key issues:

  1. It is not clear what gaps this study aims to fill. The goal of this manuscript is unclear.
  2. The discussion of the results, for the most part, simply stated what was observed with few attempts to understand the underlying mechanisms.
  3. There is no coherence in the different variables investigated. Why were these variables chosen? How do the results advance our knowledge of wax deposition mechanisms?

Below I present general and specific feedback based on the submitted manuscript.

  • Abstract: This section is poorly written, and it is difficult for a reader to follow. First, it is unclear to the reader what the study is about, and the relevance of the results presented. Also:  
    • 10-11: the first sentence is repeated
    • 16-17: In positive contrast is repeated
  • Introduction: This section is not coherent, lacks key references, and makes no compelling case for the work being presented.

To improve, I recommend:

  • A clearer presentation of why the cold finger approach is beneficial (or useful or recommended).
  • A better review of the literature to indicate what gaps this work intends to fill

Other observations (numbers refer to the lines)

  • 35: fix reference style
  • 39: What laws?
  • 39-41; Reference is missing for Hunt, Jorda, Charles
  • 41-42: Sentence is unclear
  • 44: many scholars, please include more than one reference
  • Materials and method
    • 72-74: I could not understand the steps being described
    • 75-78: Tone shifts to the imperative mood; this is unusual

  • Results and discussion:
    • 84-92: this is written in an imperative tone. It reads like it should be part of the materials and method section.
    • 116-120: this is written in an imperative tone. It reads like it should be part of the materials and method section (also applies to other sections)
    • 150-155: can the shear stresses be calculated?
    • 154: figure 5
    • 137: figure 4
    • 201-215: Figure 7 presents an interesting trend. The discussion of the results should be expanded to include more references from the literature.
  • Figures 2, 3,4,5,6,7
    • Sediment ‘quality’. Should this be quantity?
    • The plots shown are redundant. The average values could be simply shown with error bars to indicate variation in data
    • Figure 3: check the spelling on the axis and legend
    • Figure 5: x-axis should read rotational
    • Figure 4: consider plotting the deposition rates instead or in addition to.
  • Conclusion: This was divided into three. I presented my feedback below:
    • 217- 227: This section is not coherent and is difficult to follow.
    • 228-237: This simply restates some of the results previously shown.
    • 238-242: What is the basis of this recommendation?
  • Extensive editing for language is recommended: Unfortunately, there are several grammatical and presentation errors that may have undermined the science presented in this manuscript.

Author Response

Dear reviewer:

Thank you for your letter and the reviewers’ comments on our manuscript entitled "Study of wax deposition pattern of high wax-bearing crude oil based on cold finger experiment" (ID:processes-1457133). Those comments are very helpful for revising and improving our paper, as well as the important guiding significance to other research. We have studied the comments carefully and made corrections which we hope meet with approval. The main corrections are in the manuscript and the responds to the reviewers’ comments are as follows (the replies are highlighted in red ).

Point 1:10-11: the first sentence is repeated

Response 1:I deleted the duplicate sentences.

Point 2:16-17: In positive contrast is repeated

Response 2:I deleted the duplicate sentences.

Point 3:39-41; Reference is missing for Hunt, Jorda, Charles

Response 3:I removed this citation because the reference was lost.

Point 4:41-42: Sentence is unclear

Response 4:The meaning expressed in lines 41-42 of this paper is “Hamouda et al. improved the cold plate method apparatus to the cold finger method experimental apparatus, whose experimental principle is basically the same as the cold plate method , but it can better simulate the wax deposition state in the flow state, and the cold finger method was used for experimental research. 

Point 5:72-74: I could not understand the steps being described

Response 5:The meaning expressed in lines 41-42 of this paper is “Firstly, the crude oil was pretreated by heating the water bath at 60℃ for 6 hours, then the experimental temperature of the circulating water bath controller was set, and the oil sample was added after the temperature reached the experimental requirement, and the RW20 stirrer produced by IKA, Germany, was turned on to adjust the rotational speed to the required speed for the experiment. 

Point 6:75-78: Tone shifts to the imperative mood; this is unusual

Response 6:I changed "stop stirring" to “Turn off the stirrer”.

Point 7:154: figure 5  137: figure 4

Response 7:I have made changes to it.

Point 8:Figure 3: check the spelling on the axis and legend

Response 8:I have made changes to it.

Point 9:Figures 2, 3,4,5,6,7 Sediment ‘quality’. Should this be quantity?

Response 9:The meaning of the expression in the chart is the deposition mass.

Once again, thank you very much for your constructive comments and suggestions which would help us both in English and in depth to improve the quality of the paper.

Kind regards,

Da Li

Reviewer 2 Report

This manuscript presents ‘a study of Wax deposition pattern of high wax-bearing crude oil based on cold finger experiment’. This paper will provide a good solution to the current flow assurance issues. I have only minor comments on the discussion about the techniques.”

Comment 1: “From lines 10 to 12, “In order to solve the problem of pipeline wax deposition, cold fingering was used as the medium of waxy crude oil in Daqing oilfield, the factors In order to solve the problem of the pipeline wax deposition, cold fingering was used as the medium of waxy crude oil in Daqing oilfield.” There are repeated lines and it seems that the English grammar should be checked in the whole manuscript.”

Comment 2: The authors in the abstract didn’t mention a brief of the methodology and also the result is too lengthy for an abstract. The author needs to construct the abstract to be catchy to the readers.

Comment 3: “As authors indicated, pipeline transportation is widely used in practice because of its advantages such as economic and environmental protection, large transmission volume, and convenient on-site management. If possible, authors need to include the area of application to directly compare the suggested cold finger technique for wax deposition. There can be flowlines (i.e. multiphase transport), risers (multiphase transport), and export-lines or trunklines (single-phase transport) for crude oil transportation depending on its application and the technique is not expected to be the same.”

Comment 4: What is the novelty of the present work as the method of cold finger experiment has been done before by other authors? Also, I didn’t feel the connection between the literature review and the objective for this study. Please enrich these points for readability.

Comment 5: “The authors mentioned in the material and methods, the experimental setup for this experiment consisted of a cold finger, an oil tank, two circulating water bath controllers, and an additional stirring paddle. A schematic diagram of the experimental setup is shown in Figure1. Authors should furnish the readers with the following relevant information:

  1. PVT study or the compositional analysis carried out to know the components of the waxy crude as reported in the paper. There is a need to know the crude oil composition and type?
  2. The wax content determination and the Wax Appearance Temperature (WAT) for the crude oil?

Comment 6: “The authors reported Asphaltene content many times in the paper and even mentioned it as a factor for wax formation. The Authors need to carry the readers along by explaining Asphaltene, its formation, Asphaltene content determination, the relationship between Asphaltene and Wax in crude oil transport system? Because Asphaltene formation is a major issue during crude oil transportation and it is constantly been monitored to avoid its accumulation in the pipeline. Pressure has the biggest impact on asphaltene stability and Asphaltene Onset Pressure (AOP) is the pressure at which asphaltene begins to settle out in the system.

Comment 7: The Conclusion is too wordy. The Authors should improve the conclusion with fewer lines and good indentation if possible. The References section also needs to be checked for proper arrangement.

Comment 8Please carefully review the paper below for improvement on the literature. It provides a comprehensive study on the importance of flow assurance solutions to wax, asphaltene, and hydrates. Please ensure it is referenced in the literature review.

  • An Overview of Flow Assurance Heat Management Systems in Subsea Flowlines. Energies 2021, 14(2), 458; https://doi.org/10.3390/en14020458

Author Response

Dear reviewer:

Thank you for your letter and the reviewers’ comments on our manuscript entitled "Study of wax deposition pattern of high wax-bearing crude oil based on cold finger experiment" (ID:processes-1457133). Those comments are very helpful for revising and improving our paper, as well as the important guiding significance to other research. We have studied the comments carefully and made corrections which we hope meet with approval. The main corrections are in the manuscript and the responds to the reviewers’ comments are as follows (the replies are highlighted in red ).

Point 1: “From lines 10 to 12, “In order to solve the problem of pipeline wax deposition, cold fingering was used as the medium of waxy crude oil in Daqing oilfield, the factors In order to solve the problem of the pipeline wax deposition, cold fingering was used as the medium of waxy crude oil in Daqing oilfield.” There are repeated lines and it seems that the English grammar should be checked in the whole manuscript.”

Response 1: I deleted the duplicate sentences.

Point 2: The authors in the abstract didn’t mention a brief of the methodology and also the result is too lengthy for an abstract. The author needs to construct the abstract to be catchy to the readers.

Response 2: I deleted the conclusion section of the abstract and trimmed it.

Point 3:“As authors indicated, pipeline transportation is widely used in practice because of its advantages such as economic and environmental protection, large transmission volume, and convenient on-site management. If possible, authors need to include the area of application to directly compare the suggested cold finger technique for wax deposition. There can be flowlines (i.e. multiphase transport), risers (multiphase transport), and export-lines or trunklines (single-phase transport) for crude oil transportation depending on its application and the technique is not expected to be the same.”

Response 3: I have revised the question to make it clear in the text that the study is based on the trunklines pipeline.

Once again, thank you very much for your constructive comments and suggestions which would help us both in English and in depth to improve the quality of the paper.

Kind regards,

Da Li

Reviewer 3 Report

Text editing: unifying the notation of units (without spaces), for example: 51oC  not 51 o C.

Author Response

Dear reviewer:

Thank you for your letter and the reviewers’ comments on our manuscript entitled "Study of wax deposition pattern of high wax-bearing crude oil based on cold finger experiment" (ID:processes-1457133). Those comments are very helpful for revising and improving our paper, as well as the important guiding significance to other research. We have studied the comments carefully and made corrections which we hope meet with approval. The main corrections are in the manuscript and the responds to the reviewers’ comments are as follows (the replies are highlighted in red ).

Point 1:Text editing: unifying the notation of units (without spaces), for example: 51oC  not 51 C.

Response 1:Changes have been made to address this issue.

Once again, thank you very much for your constructive comments and suggestions which would help us both in English and in depth to improve the quality of the paper.

Kind regards,

Da Li

Reviewer 4 Report

In the abstract the term 90r/min is not defined, and is not clear until far into the manuscript. There is also no "law"  of wax deposition. It is a physical process that can be measured for various experimental conditions, but a "law" does not exist.

Figure 1 should show arrows indicating fluid flow. Water is the correct spelling, not watrer.

Figure 2-7. The Y-axis should be sediment quantity, not sediment quality.

Line 116-120. This discussion should be moved below Figure 3.  This comment also applies to lines 132-137 and beyond for the additional figures. The discussion of a figure should be below the figure.

Line 183-191 Define "silt." It is not clear if this is loose and floating particulate or wax adhered to the wall of the cold finger.

There is inconsistent use of upper case letters in the references. For example, in References 6 and 7, there has been upper case letters used for the surname of Chinese authors. Normally, upper case is used for initials and only the first letter of the surname is upper case.   For example, Wang is preferred over WANG.

In the Methods and Materials section, perhaps inserting a table that lists the experimental conditions that will subsequently carried out and the results presented in Figures 2, 3, 4, 5, 6, 7. This will allow elimination of repetitious discussion introducing each of Figures 2 through 7.

Author Response

Dear reviewer:

Thank you for your letter and the reviewers’ comments on our manuscript entitled "Study of wax deposition pattern of high wax-bearing crude oil based on cold finger experiment" (ID:processes-1457133). Those comments are very helpful for revising and improving our paper, as well as the important guiding significance to other research. We have studied the comments carefully and made corrections which we hope meet with approval. The main corrections are in the manuscript and the responds to the reviewers’ comments are as follows (the replies are highlighted in red ).

Point 1: Figure 1 should show arrows indicating fluid flow. Water is the correct spelling, not watrer.

Response 1:Modifications have been made to Figure 1.

Point 2: Figure 2-7. The Y-axis should be sediment quantity, not sediment quality.

Response 2:In response to this issue, “quality” has been changed to “quantity”.

Point 3: Line 116-120. This discussion should be moved below Figure 3.  This comment also applies to lines 132-137 and beyond for the additional figures. The discussion of a figure should be below the figure.

Response 3:Changes were made to address this issue.

Point 4: Line 183-191 Define "silt." It is not clear if this is loose and floating particulate or wax adhered to the wall of the cold finger.

Response 4:In response to this issue, “silt” has been changed to “sediment”.

Point 5: There is inconsistent use of upper case letters in the references. For example, in References 6 and 7, there has been upper case letters used for the surname of Chinese authors. Normally, upper case is used for initials and only the first letter of the surname is upper case.   For example, Wang is preferred over WANG.

Response 5:Changes were made to address this issue.

Once again, thank you very much for your constructive comments and suggestions which would help us both in English and in depth to improve the quality of the paper.

Kind regards,

Da Li

Round 2

Reviewer 1 Report

Dear authors,

I have checked the updated manuscript against comments received during the first round of review. While some minor changes were made to the text, for example, changing ‘silt’ to ‘sediment’, it appears that several major areas that were pointed out in the first review, have not been addressed.

Below, I post key comments from the previous round that were not addressed by the authors in the current version:

  • Introduction: This section is not coherent, lacks key references, and makes no compelling case for the work being presented.

To improve, I recommend:

  • A clearer presentation of why the cold finger approach is beneficial (or useful or recommended).
  • A better review of the literature to indicate what gaps this work intends to fill
  • Results and discussion:
    • 84-92: this is written in an imperative tone. It reads like it should be part of the materials and method section.
    • 116-120: this is written in an imperative tone. It reads like it should be part of the materials and method section (also applies to other sections)
    • 150-155: can the shear stresses be calculated?
    • 201-215: Figure 7 presents an interesting trend. The discussion of the results should be expanded to include more references from the literature.
  • Conclusion: This was divided into three. I presented my feedback below:
    • 217- 227: This section is not coherent and is difficult to follow.
    • 228-237: This simply restates some of the results previously shown.
    • 238-242: What is the basis of this recommendation?

Extensive editing for language is recommended: Unfortunately, there are several grammatical and presentation errors that may have undermined the science presented in this manuscript. 

Author Response

Dear reviewer:

Thank you for your letter and the reviewers’ comments on our manuscript entitled "Study of wax deposition pattern of high wax-bearing crude oil based on cold finger experiment" (ID:processes-1457133). Those comments are very helpful for revising and improving our paper, as well as the important guiding significance to other research. We have studied the comments carefully and made corrections which we hope meet with approval. The main corrections are in the manuscript and the responds to the reviewers’ comments are as follows (the replies are highlighted in red ).

Point 1: This section is not coherent, lacks key references, and makes no compelling case for the work being presented. To improve, I recommend: A clearer presentation of why the cold finger approach is beneficial (or useful or recommended). A better review of the literature to indicate what gaps this work intends to fill.

Response 1: In response to recommendation 1, I made changes that “I deleted the duplicate sentences.Due to the high cost of the experimental loop, the experimental cost is large and the thickness of the sediment cannot be directly obtained. “ In response to recommendation 2, I wrote in the first paragraph to find the law of pipe deposition and solve the safety problems caused by pipe condensation, etc.

Point 2: 84-92: this is written in an imperative tone. It reads like it should be part of the materials and method section.

Response 2: I have placed this section and the same section that follows at the bottom of the image, and this paragraph shows the experimental conditions set according to the relevant influencing factors.

Point 3: 150-155: can the shear stresses be calculated?

Response 3: I read a lot of references and did not find a relevant formula for calculating the shear rate.

Point 4: 150-155: can the shear stresses be calculated?

Response 4: I read a lot of references and did not find a relevant formula for calculating the shear rate.

Point 5: 

  • 217- 227: This section is not coherent and is difficult to follow.
  • 228-237: This simply restates some of the results previously shown.

Response 5: Paragraphs 1 and 2 of the conclusion are the results obtained from the above experiments.

Once again, thank you very much for your constructive comments and suggestions which would help us both in English and in depth to improve the quality of the paper.

Kind regards,

Da Li

Reviewer 4 Report

In the next revision, place the figures in the proper position and delete the old obsolete figures.

The Conclusion needs to be completely re-written. The Conclusion might contain more qualitative statements about the effects of temperature, temperature differences between the wall and the oil, rotation, etc. Do not repeat the numeric data that is in the main text and in the figures.

Author Response

Dear reviewer:

Thank you for your letter and the reviewers’ comments on our manuscript entitled "Study of wax deposition pattern of high wax-bearing crude oil based on cold finger experiment" (ID:processes-1457133). Those comments are very helpful for revising and improving our paper, as well as the important guiding significance to other research. We have studied the comments carefully and made corrections which we hope meet with approval. The main corrections are in the manuscript and the responds to the reviewers’ comments are as follows (the replies are highlighted in red ).

Point 1: 

In the next revision, place the figures in the proper position and delete the old obsolete figures.

The Conclusion needs to be completely re-written. The Conclusion might contain more qualitative statements about the effects of temperature, temperature differences between the wall and the oil, rotation, etc. Do not repeat the numeric data that is in the main text and in the figures.

Response 1: Changes have been made to address this issue.

Once again, thank you very much for your constructive comments and suggestions which would help us both in English and in depth to improve the quality of the paper.

Kind regards,

Da Li